# Accurate Recapitulation of Chikungunya Virus Complete Coding Sequence Phylogeny Using Variable Genome Regions for Genomic Surveillance

**DOI:** 10.3390/v16060926

**Published:** 2024-06-07

**Authors:** Eduardo D. Rodríguez-Aguilar, Everardo Gutiérrez-Millán, Mario H. Rodríguez

**Affiliations:** Center for Infectious Disease Research, National Institute of Public Health of Mexico, Av. Universidad 655, Cuernavaca 62100, Mexico; rodriguezaed@gmail.com (E.D.R.-A.); ever.gmillan@gmail.com (E.G.-M.)

**Keywords:** Chikungunya virus, phylogeny, genomic surveillance

## Abstract

Chikungunya virus (CHIKV) is transmitted by mosquito bites and causes chikungunya fever (CHIKF). CHIKV has a single-stranded RNA genome and belongs to a single serotype with three genotypes. The Asian lineage has recently emerged in the Western Hemisphere, likely due to travel-associated introduction. Genetic variation accumulates in the CHIKV genome as the virus replicates, creating new lineages. Whole genome sequencing is ideal for studying virus evolution and spread but is expensive and complex. This study investigated whether specific, highly variable regions of the CHIKV genome could recapitulate the phylogeny obtained with a complete coding sequence (CDS). Our results revealed that concatenated highly variable regions accurately reconstructed CHIKV phylogeny, exhibiting statistically indistinguishable branch lengths and tree confidence compared to CDS. In addition, these regions adequately inferred the evolutionary relationships among CHIKV isolates from the American outbreak with similar results to the CDS. This finding suggests that highly variable regions can effectively capture the evolutionary relationships among CHIKV isolates, offering a simpler approach for future studies. This approach could be particularly valuable for large-scale surveillance efforts.

## 1. Introduction

Chikungunya virus (CHIKV) is an alphavirus transmitted by the bite of infected *Aedes aegypti* and *Aedes albopictus* female mosquitoes. It causes chikungunya fever (CHIKF), an acute febrile disease with four clinical forms: acute, atypical acute, severe acute and chronic [1]. The CHIKV genome consists of a single stranded positive sense 12 kB RNA. The viral mRNA includes a 5′ methylguanylate cap and a 3′ polyadenylated tail, encoding two open reading frames (ORFs) for non-structural and structural polyproteins [2]. The non-structural ORF encodes four proteins (nsp1, nsp2, nsp3, and nsp4) essential for transcription, replication and caping of the viral RNA, and the viral polyprotein cleavage; the structural ORF encodes five proteins (C, E3, E2, 6K, y E1) [3].

CHIKV comprises a single serotype [4], with three genotypes distinguished by their historical geographic distributions: West African (WAf), East/Central/South African (ECSA), and Asian lineages [5,6]. In 2005–2006, the Indian Ocean Lineage (IOL), a strain of the ECSA lineage, sparked a significant epidemic in the Indian Ocean islands and the Indian subcontinent, marking the first major outbreak after 32 years of inactivity [5,7]. Isolated cases, associated with travelers returning from this region, led to small outbreaks in Italy in 2007 [8] and France in 2010 [9]. 

It has been suggested that the emergence of CHIKV in the Western hemisphere dates back to the 19th century, with cases potentially misdiagnosed as dengue fever (DENF) [10]. This speculation is plausible due to the overlapping signs and symptoms of CHIKF and DENF [11]. Nonetheless, before December 2013, when a CHIKF outbreak was reported in the Caribbean Island of Saint Martin, there was no substantial evidence of local transmission [12].

The immunologically naive population in the Americas, the widespread *Ae. aegypti* mosquitoes, and the frequent travel between islands facilitated the rapid spread of the virus throughout the Caribbean and eventually to the American mainland [13]. By February 2014, autochthonous transmission had been confirmed in ten territories, leading to 2582 confirmed cases of CHIKF [14]. By December 2014, transmission had been confirmed in 33 territories, with a total of 1,071,696 suspected and 22,796 confirmed cases [14]. Certain strains of the IOL undergo adaptive mutations in the E1 and E2 envelope glycoproteins, enhancing its infectivity for *Ae. albopictus* [15]. This adaptation opened opportunities for CHIKV transmission in regions where this mosquito is prevalent and its extension beyond the tropical and subtropical areas where *Ae. aegypti* is endemic [16]. 

The significant number of infected travelers between 2006 and 2008 raised expectations that the IOL strain would be the first to emerge in the Western hemisphere [17]. However, sequence analysis has revealed that CHIKV isolated from Saint Martin at the onset of the outbreak belonged to the Asian lineage [12]. Subsequent analysis of CHIKV from the Caribbean islands, Martinique [18], the British Virgin Islands [19], and Trinidad [20], as well as from the American mainland, including Mexico [21], Panama [22],Colombia [23], and Brazil [24], has confirmed the presence of the Asian lineage. On the other hand, alongside the circulation of the Asian genotype, an outbreak caused by a newly introduced ECSA strain occurred in the east-central region of Brazil [24]. As sequence data were limited to a small subset of CHIKV-affected countries, it remains uncertain whether the ECSA genotype was more prevalent, other genotypes were circulating, or if the outbreak stemmed from a single Asian genotype introduction. Analyzing genomic sequences from a broader region could enhance accuracy in estimating evolutionary rates and identifying the selective pressures influencing viral evolution.

The spread of CHIKV is accompanied by the accumulation of genetic variation, leading to the emergence of distinct lineages with potential virulence and transmission differences [25]. The accumulation of genetic variations is mainly produced by errors generated by RNA polymerase during viral replication (10^3^ to 10^5^ substitutions per nucleotide copied per round of replication) [26]. Although mutations arise stochastically within the genome, viral genes contain conserved regions, critical to preserve structure and function, combined with regions permissive to multiple mutations, which allow immune evasion through antigenic variation [27]. 

Advances in DNA sequencing and bioinformatics tools have improved viral genomic surveillance [28]. These allow thorough examination of the genome and dissemination patterns of emerging and reemerging pathogens [29]. Despite this progress and the significant number of reported CHIKF cases in recent years, public genomic data primarily consists of genomes from localized outbreaks, limiting their representativeness [30]. Extensive genomic surveillance efforts can provide insights into the genetic diversity of the circulating viral populations during outbreaks and periods of reduced transmission. However, epidemiological surveillance based on whole genomes is limited in many endemic countries due to the technical and analytical complexity and high costs [31]. The inherent variability within the CHIKV genome presents an opportunity to identify hypervariable regions. If these regions demonstrate sufficient phylogenetic signal to recapitulate results obtained from complete genome analysis, they could serve as useful alternatives for evolutionary studies.

Previously, we demonstrated that highly variable regions of the dengue virus genome can be useful in recapitulating the phylogeny obtained using the whole genome [32]. In this study, we examined genetic variability across the complete coding sequence (CDS) of CHIKV of a selection of sequences from around the world. We examined the performance of various genomic regions with differing variability levels for phylogenetic analysis, compared to that obtained with the whole genome. We propose that hypervariable genomic regions could be useful for monitoring CHIKV evolution through ongoing genomic surveillance efforts.

## 2. Materials and Methods

### 2.1. Genetic Variability Analysis

A total of 1114 CHIKV complete genome sequences from the Bacterial and Viral Bioinformatics Resource Center (BV-BRC) of the National Institute of Allergy and Infectious Diseases (NIH/DHHS), available at https://www.bv-brc.org/ (accessed on 5 February 2024), were included in this study.

We conducted a genetic variability analysis to explore whether specific regions of the CHIKV genome (conserved or variable regions) could yield useful phylogenetic signals [32]. We identified candidate regions and assessed their ability to replicate the phylogeny compared to that using the complete coding sequence (CDS). To identify conserved and highly variable regions, we aligned 45 nt non-overlapping sliding windows of the CDS sequences using Clustal Omega software (v. 1.2.1) [33]. For each window, we calculated the number of nucleotide changes and the ratio of synonymous to non-synonymous mutations for every codon. Three regions exhibiting the highest number of mutations and the highest ratios of synonymous to non-synonymous mutations, as well as three regions displaying the lowest number of mutations and the lowest rates of synonymous to non-synonymous mutations, were selected for subsequent phylogenetic analysis.

### 2.2. Phylogenetic Analysis

To assess the sensitivity of various regions in discriminating between sequences from different lineages and geographic origins, datasets were constructed with sequences from Africa, Asia, and America. These datasets incorporated sequences belonging to the three well-defined genotypes (Asian and Caribbean, ECSA, and West Africa) and lineages such as IOL and the Caribbean. Redundant sequences were removed from the initial datasets to prevent overrepresentation caused by oversampling. The remaining sequences were then randomly chosen to create six different alignments, each consisting of 30 sequences (detailed sequence data from the six alignments are presented in Appendix A).

Potential sequence recombination events were screened using Genetic Algorithm for Recombination Detection (GARD), accessible via the Datamonkey web server, available at https://www.datamonkey.org/gard (accessed on 26 February 2024). Multiple sequence alignments for each alignment, based on the Percent Nucleotide Identities (PNI) calculated using p-distances, were generated using Clustal Omega (Version 1.2.1) [33]. From these alignments, three highly variable regions (Hi) of 100, 300, 500, 700, 900, 1200, 1500, and 2100 nt each, corresponding to regions of genes E2, NSP1, and NSP3 (Hi-E2, Hi-NSP1, and Hi-NSP3), and three regions with a low rate of variability (Lo) of 100, 300, 500, 700, 900, 1200, 1500, and 2100 nt each, corresponding to regions of genes C, NSP1, and NSP2 (Lo-C, Lo-NSP1, and Lo-NSP2) were extracted. 

Separate phylogenetic analyses were performed using the three highly variable regions of identical lengths, as well as concatenated sequences constructed with the three high (Hi-Concatenated) or low (Lo-Concatenated) variable regions, respectively (3 × 100, 3 × 300, 3 × 500, and 3 × 700 nt). The best-fit model of nucleotide substitution for each alignment was determined based on the Bayesian information criterion (BIC) using ModelTest 3.5 [34]. The GTR + G + I model (general time-reversible model with gamma-distributed rates of variation among sites and a proportion of invariable sites) was identified as the best-fit model for all alignments. Phylogenetic trees were constructed with these sequences using Mr. Bayes software v.3.2 [35].

### 2.3. Recapitulating the Phylogeny of the CDS

Using trees constructed with the complete coding region (CDS) of each alignment as references, we assessed the performance of each candidate sequence (identified from the genetic variability analysis) in reproducing the phylogeny obtained with the entire coding sequence. For each tree constructed with the different regions under evaluation, we computed relative branch length, topological incongruence, and tree confidence. Relative branch length was determined using the branch length distance (BLD). The BLD metrics quantify the similarity in branch lengths between phylogenetic trees. They calculate a scaling factor that minimizes the total evolutionary divergence between a reference tree and a compared tree. A BLD value of 1 indicates identical total branch length in both trees. Values greater than 1 suggest that the compared tree has proportionally shorter branches compared to the reference one, while values less than 1 indicate proportionally longer branches, thus quantifying the scaling factor to approximate the overall divergence between the reference and compared trees [36]. Topological incongruences were evaluated using the K-tree score, measuring the minimum branch length distance between trees after scaling. Lower K-tree scores indicate greater similarity to the reference tree. This calculation was performed using the Perl program Ktreedist Version 1.0 [37]. Additionally, tree confidence was computed as the mean posterior probability (mean PP) [38]. This parameter was normalized based on the number of nodes in each group of trees resulting from summing the posterior probability of all nodes in each compared tree divided by the number of nodes in the corresponding reference tree.

## 3. Results

### 3.1. Genetic Variability Analysis

The analysis of CHIKV genome variability identified regions of high variability and highly conserved regions (Figure 1).

Analysis of the hypervariable regions and the proportion of synonymous and non-synonymous mutations showed that the region with the highest number of mutations occurred in the NSP3 gene, followed by regions in the NSP1, NSP4, and E2 genes. Analysis of synonymous and non-synonymous mutations across the CDS showed a pattern of peaks with a high proportion of non-synonymous mutations alternating with peaks with a low proportion of non-synonymous mutations. The highest peak of non-synonymous mutations (nearly 100% of the mutations found were non-synonymous) occurred in the NSP1 gene. The NSP3 and E2 genes showed regions with peaks over 50% of non-synonymous mutations. The regions with the lowest number of mutations and the lowest rate of synonymous and non-synonymous mutations were in NSP1, NSP2, and C (including regions where no non-synonymous mutations were found) (Figure 1).

To assess the capacity of regions with genetic variation in recapitulating the CDS phylogeny, we used the three regions of the genome with the highest numbers of mutations and the highest proportions of non-synonymous mutations, separately, and a sequence formed by concatenating the same three regions. These included regions of the NSP1, NSP3, and E2 genes. The three regions of the genome exhibiting minimal mutational burden and the lowest proportions of non-synonymous mutations, isolated and concatenated, were used as controls.

### 3.2. Evaluation of the Suitability of Hypervariable Regions for Recapitulating CDS Phylogeny

#### 3.2.1. Branch Length Distance

Analysis of BLD revealed that regions of both high and low variability up to 700 nucleotides (100, 300, 500, and 700 nt) resulted in scaling factors surpassing 1. This trend remained consistent across all regions, except for Hi-NSP3, which consistently generated scaling factors below 1. These results imply that phylogenetic trees constructed from these variable regions display relatively short branches compared to trees reconstructed using the CDS (Figure 2). For regions of 900 nt or longer (1200, 1500, and 2100 nt), BLD values progressively approached 1, reaching a point where no statistically significant differences were observed compared to an ideal scaling factor. This suggests that longer highly variable regions may more accurately reflect viral phylogeny compared to the shorter regions. Concatenated highly variable regions maintained values below 1, except for the 2100 nt region. The low-variability regions did not follow this trend, consistently exhibiting BLD values above 1 across all analyzed lengths (Figure 2).

#### 3.2.2. Topological Incongruence

Topological incongruence between phylogenetic trees was assessed using the K-tree score. This metric calculates the minimum branch length distance required to transform one tree into another after scaling them to the CDS reference tree length. Across all analyzed regions, there was a consistent decline in K-tree scores as the length of the analyzed region increased. For 100 nt regions, K-tree scores varied from 0.03 to 0.15, indicating a high level of uncertainty in the tree’s reconstruction. This variability slightly decreased for 300 nt regions, with scores ranging from 0.03 to 0.08. A notable enhancement was observed as the analyzed region size increased. For 2100 nt regions, k-tree scores exhibited a much narrower range, with NSP3 being the best-performing region, with scores between 0.008 and 0.01. In contrast, concatenated low-variability regions showed the lowest performance, having scores ranging from 0.009 to 0.03. In contrast, the high-variability and the low-variability 100, 300, 1500, and 2100 nt regions of NSP3 presented statistically significant differences. In addition, reduced dispersion in K scores was observed with increasing region size, indicating that larger regions provide a more robust signal for reconstructing the evolutionary relationships represented by the topology of CSD trees (Figure 3).

#### 3.2.3. Tree Confidence

Tree confidence, assessed using mean posterior probability (mean PP), required normalization due to variations in the number of nodes recovered across different tree sets. The CDS phylogenetic tree exhibited high confidence in its branching patterns. This is reflected in the mean PP values for the nodes within the tree, which ranged from 0.94 to 1. Mean PP values from the high- and low-variability regions exhibited a consistent increase as the analyzed region length increased and regions with high variability performed better than regions with low variability. Analysis of nucleotide region length revealed a consistent positive correlation between region length and mean posterior probability (PP) values for both high- and low-variability regions. High-variability regions exhibited consistently higher mean PP values compared to their low-variability counterparts. For instance, at a region length of 100 nucleotides, the mean PP values in high-variability regions ranged from 0.23 to 0.48, while those in low-variability regions spanned only 0.04 to 0.17. Despite initially exhibiting very low mean PP values, high-variability regions demonstrated a significant increase in tree confidence, reaching moderate levels (mean PP values greater than 0.75) by 1200 nucleotides (mean PP: 0.62–0.82). Also, very high confidence consistency (mean PP values greater than 0.9) was only achieved with high-variability regions Hi-NSP3 and HiE2, and the concatenated high-variability region, all analyzed at region lengths of 2100 nucleotides. This result suggests that larger regions provide a more robust phylogenetic signal, ultimately enhancing the confidence of reconstructed trees. In contrast, concatenated low-variability regions exhibited statistically significant decreases in mean PP values compared to all other regions (regardless of length). Conversely, concatenated highly variable regions demonstrated better mean PP values compared to other regions, ultimately reaching a point where no statistically significant differences were observed from the values obtained using the reference tree (CSD) (Figure 4). (Detailed pairwise statistical comparisons of scaling factors, K-scores, and mean PP values between all analyzed regions are provided in Appendix A).

### 3.3. Inferring the Evolutionary Relationships among CHIKV Isolates from the American Outbreak Using Genomic Regions of Different Variability

To evaluate the suitability of highly variable genome regions of CHIKV for outbreak tracking through genomic surveillance, we employed a comparative phylogenetic approach. We reconstructed phylogenetic trees using the CDS of CHIKV isolates from the three distinct genotypes (Asian and Caribbean, ECSA, and West Africa) previously determined using the complete genome and sequences collected during the American outbreak (detailed sequence data employed for this analysis can be found in Appendix A). Subsequently, we generated phylogenetic trees for the same isolates using the selected genomic regions exhibiting varying degrees of variability. By comparing the resulting trees to the reference trees derived from the complete CDS, we assessed the congruence between the evolutionary relationships inferred from these distinct genomic regions. The phylogenetic tree constructed using the CDS effectively distinguished the three distinct genotypes (Asian and Caribbean, ECSA, and West Africa). Within the ECSA genotype, it further differentiated between the lineage circulating in Asia and the one that circulated in Brazil during the 2014 outbreak. Additionally, within the Asian and Caribbean genotype, it distinguished between the lineages circulating before the main inland American outbreak. The Asian sequences from 2007 and 2012 were correctly separated from the 2013 sequences, which are hypothesized to carry the adaptations responsible for the American outbreak. The sequences of American isolates also formed a distinct clade, including the sequence isolated in St. Martin at the beginning of the outbreak (Figure 5a).

The ability to differentiate the viral lineages of the different regions evaluated (all of 2100 nt length) varied. While all regions successfully distinguished the three major genotypes (Asian and Caribbean, ECSA, and West Africa), their discriminatory power varied when resolving finer-scale relationships within genotypes. Notably, Hi-NSP1 and the concatenated highly variable regions provided sufficient phylogenetic signal to differentiate the lineages within the Asian and Caribbean genotype (2007, 2012, 2013 Asian sequences, and American sequences) (Figure 5b,c). The Low-NSP1 region failed to differentiate the 2012 Asian sequences from the 2013 sequences preceding the American outbreak. Low-C, Low NSP-2, and concatenated low-variability regions grouped the pre-outbreak Asian sequences with the outbreak isolates in a single clade. Similarly, the Hi-E2 region could not distinguish the 2007 from the 2012 Asian sequences, and the Hi-NSP3 region did not differentiate the pre-outbreak Asian sequences from the American outbreak isolates (Appendix A). 

## 4. Discussion

Chikungunya virus exhibits a remarkably rapid evolutionary rate, with mutations accumulating at a pace that closely aligns with its rapid transmission dynamics [39]. This translates to detectable genetic differences between lineages responsible for epidemics emerging within a short period, often on the scale of months [21,40]. Whole-genome sequencing (WGS) offers the most comprehensive view of CHIKV genetic variation, allowing for the identification of even minor mutations that might influence viral characteristics [41]. Although whole-genome sequencing is becoming more accessible, its costs are still a major barrier. Additionally, processing and analyzing the massive datasets requires sophisticated infrastructure, which is often lacking in resource-constrained settings [42]. NGS platforms hold considerable potential for chikungunya genome surveillance programs, enabling detailed characterization of viral strains. However, despite significant advancements in sequencing speed and technology, the cost of establishing and running a dedicated NGS facility [31] remains a substantial challenge for many chikungunya surveillance programs, particularly in low- and middle-income countries [31]. Additionally, individual sequencing runs incur costs depending on the type of analysis. For example, whole-genome sequencing can range from USD 100 to USD 1000 per sample based on the latest pricing trends [42]. Oxford Nanopore Technologies provides a more accessible and affordable form of genome sequencing [43], although with an error rate of around 14% [44]. This high error rate could hinder accurate assembly due to fragmented reads and ambiguous base calls, which could compromise downstream analyses, leading to misinterpretations of the viral sequence, and hinder understanding of the molecular epidemiology and evolutionary patterns of CHIKV [45]. This study proposes the use of highly variable genomic regions as a substitute for CDS. These strategically selected regions can effectively reproduce phylogenies derived from whole genomes and could be used, at lower cost, using PCR methods, for monitoring CHIKV evolution through ongoing genomic surveillance efforts. The PCR conditions and primers proposed for NSP1, NSP3, and E2 partial gene amplification are presented in Appendix A.

The selection of appropriate genomic loci remains a crucial step in constructing robust phylogenetic trees, particularly when aiming at resolving relationships at different evolutionary scales. Evolutionary rate is widely recognized as a critical factor influencing the usefulness of any chosen locus [46,47,48]. The standard approach of studying molecular diversification using only a single structural gene of the virus [49,50] has been applied to CHIKV. However, the limited phylogenetic signal generated incongruent phylogenetic hypotheses lacking enough support [51]. For instance, a maximum likelihood analysis of the E1 gene failed to resolve the ECSA lineage as monophyletic [52], as demonstrated in whole-genome analyses and our findings. Genes evolving at a slow pace, compared to those of divergent lineages represented in a phylogenetic tree, may accumulate too few mutations to accurately resolve their evolutionary history.

The analysis of untranslated regions (UTRs) could greatly benefit the study of short-term adaptations in RNA viruses. These regions evolve rapidly, providing valuable insights [53]. However, this approach requires CDS sequencing, and presents significant challenges, because complex structures, low GC content, and repetitive elements can lead to errors during sequencing [54]. In addition, excessive rapid evolution can lead to a phenomenon known as saturation, where independent mutations obscure the true evolutionary history at a particular site [55]. This effect reduces the discriminatory power of the locus, hindering the ability to distinguish between closely related lineages [56]. Our study revealed different rates of nucleotide variation within the CHIKV genome, characterized by both highly variable and highly conserved regions. Notably, regions in genes encoding the NSP1, NSP3, and E2 proteins exhibited the highest degree of variability, potentially reflecting their roles in functions less constrained by selective pressure. Conversely, genome regions encoding NSP1, NSP2, and C proteins displayed the lowest variability, suggesting they might be under strong purifying selection to maintain essential viral functions.

Longer branches of phylogenetic trees indicate a greater number of mutations accumulated along the biological story of each lineage [48]. Our phylogenetic trees constructed using highly variable regions displayed longer branches compared to those generated with less variable regions [57]. However, the length of the genomic sequences employed in the analysis was important for the estimation of the accumulated variations, and trees built with short, highly variable regions did not accurately reflect distances among lineages. In this case, all evaluated regions, except for the highly variable NSP3 region (Hi-NSP3), resulted in shorter branches compared to the reference tree. The NS1 gene harbors a region with the highest combined rate of synonymous and non-synonymous mutations. However, the NSP3 gene exhibits the highest overall number of mutations, including a substantial proportion of both synonymous and non-synonymous changes. This pattern suggests a high degree of positive selection acting on the NSP3 region. Positive selection describes a process where mutations that confer an evolutionary advantage become more frequent over time within a population. In the context of viruses, these advantageous mutations might enhance essential functions for viral replication, transmission, or immune evasion [58]. For RNA viruses like CHIKV, with limited coding capacity, mutations can have a significant effect on protein structure and function [27]. Regions under positive selection, like the NSP3 gene in this case, might experience a higher proportion of non-synonymous mutations. These functionally relevant mutations likely contribute to an accelerated evolutionary rate in the NSP3 gene relative to other regions. This phenomenon, driven by positive selection, can lead to inflated mutation rates in specific regions. Consequently, branch lengths in phylogenetic trees may not accurately reflect the true evolutionary distances, leading to an overestimation of genetic distances between viral isolates. Our study suggests that using longer genomic regions or combining regions with varying levels of variability can mitigate this issue. The observed improvement with longer regions might be due to the mosaic nature of the CHIKV genome, where highly variable and conserved regions are interspersed. By increasing the analyzed region size, we captured a broader spectrum of the variability within the genome. This broader sampling provides a more accurate representation of the overall evolutionary rate across the CDS.

In contrast, low-variability regions exhibited a significantly diminished capacity to reproduce the CSD phylogeny branch length. Short branches potentially indicate a slower evolutionary rate within these regions, resulting in a limited number of accumulated mutations [59]. In the context of phylogenetic analysis, such limited variation presents a challenge. These regions might lack the necessary informative sites to accurately resolve relationships between lineages [60]. The limitations of low-variability regions in reconstructing the true evolutionary history (branch lengths and divergence times; BLD) likely influenced the performance in other evaluated parameters. This became evident in two key observations: First, shorter regions, which inherently captured less sequence variation, and yielded low K-tree score values. Low scores in this case suggest that the short, less variable regions lack sufficient informative sites to accurately reconstruct branch lengths, potentially misleading estimates of evolutionary distances between lineages. Second, the mean posterior probability values, which measure the level of tree confidence, were consistently low across all region lengths. This determines that low-variability regions provide weak phylogenetic signals, making it difficult to confidently resolve relationships between lineages and unreliable tree inferences [38]. In contrast, highly variable regions contributed to more robust inferences across all evaluated parameters, showcasing their effectiveness in reconstructing CHIKV phylogenies. The concatenated highly variable regions achieved a level of accuracy statistically equivalent to the reference CDS phylogeny. This performance can be attributed to the lower susceptibility to substitution saturation of highly variable regions. Each gene region provides a unique evolutionary perspective, and combining this information increased the overall number of informative sites for analysis [61].

The capacity to distinguish between closely related CHIKV strains is vital for genomic surveillance. Genome surveillance programs provide comprehensive insights into the genetic makeup and evolutionary dynamics of pathogens [62]. This approach allows for a better understanding of disease spreading, aiding in the development of targeted prevention and control strategies [63,64]. Concatenated highly variable regions demonstrated remarkable performance in reconstructing the CSD phylogeny. Using a region size of 2100 nucleotides, the K-tree scores fell below 0.01, with no statistically significant differences to the reference CSD phylogeny. Concatenated highly variable regions of this length offered a robust approach for phylogenetic analysis.

To illustrate the potential of this approach, we investigated the applicability of 2100 nt regions for genomic analysis during the last outbreak in the Americas. For this, sequences from the three different CHIKV genotypes were chosen, including the Asian genotype and the Caribbean lineage, since distinguishing between closely related sequences during the outbreak was the main challenge in this analysis. The lineage responsible for the American outbreak is hypothesized to have undergone adaptive mutations in Asia, enabling efficient spread through *Ae. aegypti* [29], years before the outbreak itself. However, detecting these subtle mutations is vital for early intervention. All analyzed 2100-nucleotide regions successfully differentiated between the major genotypes (Asian/Caribbean, ECSA, West Africa), probably because of their substantial genetic distances. In addition, the Hi-NSP1 region and concatenated highly variable regions provided sufficient signal to differentiate lineages within the Asian/Caribbean genotypes. These regions, by distinguishing pre-outbreak Asian sequences from those isolated during the outbreak, captured more informative evolutionary changes compared to the others. By distinguishing pre-outbreak Asian sequences from those isolated during the outbreak, these regions offered valuable insights into the evolutionary trajectory of the virus and the emergence of potentially more transmissible variants.

Our study revealed that 2100 concatenated highly variable regions could be used to recapitulate CDS phylogenetic trees, but those shorter than 1500 nucleotides lacked sufficient resolution to decipher the intricate relationships between viral strains within the American outbreak. Focusing on informative regions with adequate length, like Hi-NSP1 or concatenated highly variable regions, can provide a balance between feasibility and detecting subtle evolutionary changes crucial for monitoring transmission patterns, identifying variants, and informing public health interventions. Future research could explore additional targeted regions and optimal combinations for concatenated approaches to maximize resolution while maintaining feasibility.

## Figures and Tables

**Figure 1 viruses-16-00926-f001:**
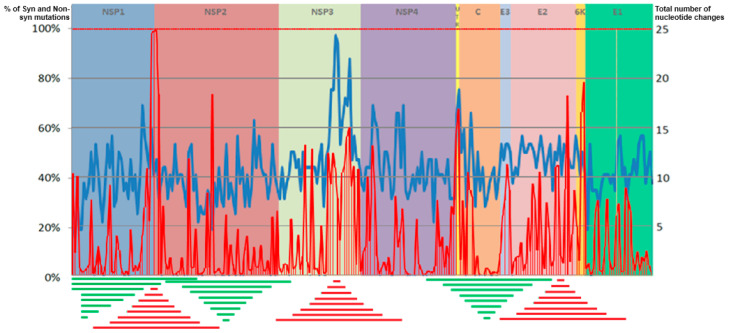
Genetic variability analysis of the CHIKV complete coding sequence (CSD). The analysis utilized 924 complete CHIKV genome sequences from the Bacterial and Viral Bioinformatics Resource Center (BV-BRC). A recurring pattern of peaks and troughs across the CDS reflect variations in the mutation rate and the relative proportions of synonymous and non-synonymous nucleotide substitutions. The *x*-axis corresponds to the nucleotide position within the CDS. The continuous blue line (right *y*-axis) represents the total number of nucleotide changes observed within a 45-nucleotide sliding window. The red line depicts the percentage of synonymous and non-synonymous mutations (left *y*-axis). The lengths of the regions used in subsequent analyses are depicted below the figure. High-variability regions (red lines) correspond to nucleotide numbers 100, 300, 500, 700, 900, 1200, and 1500 nt. Low-variability regions (green lines) correspond to positions 2100, 1500, 900, 700, 500, 300, and 100 nt.

**Figure 2 viruses-16-00926-f002:**
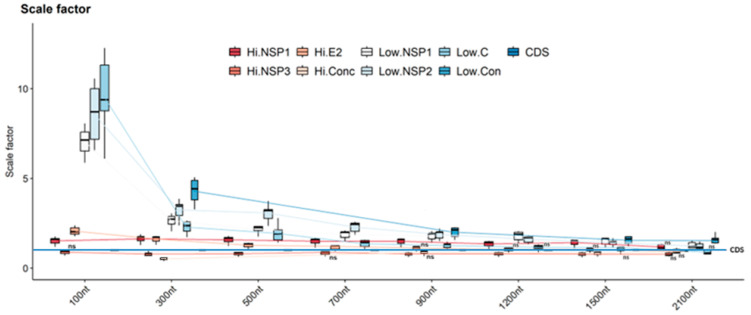
Comparison of relative branch length between CDS tree and trees constructed with high- and low-variation sequences. The trees constructed with the different regions (Hi- and Low-variables) and different lengths (100, 300, 500, 700, 900, 1200, 1500, and 2100 nt) were compared with the one constructed with their respective CDS to obtain the scaling factor. For each column, the box represents the second (Q2) and third (Q3) quartiles of the data distribution, the line within the box marks the median, whiskers denote the most extreme data. One-way ANOVA analysis of variance and the *p*-value of Tukey’s test were used to identify specific regions with non-significant differences compared to the CDS phylogeny. Regions with *p*-values “ns” (not significant) denote *p* values greater than 0.05.

**Figure 3 viruses-16-00926-f003:**
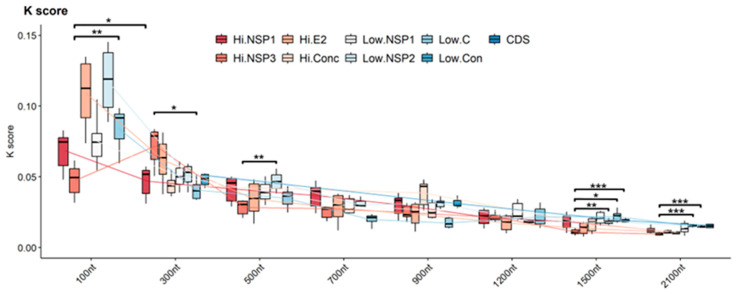
Comparison of the topological incongruence between CDS tree and trees constructed with high- and low-variation sequences. The trees constructed with the different regions (Hi- and Low-variables) and different lengths (100, 300, 500, 700, 900, 1200, 1500, and 2100 nt) were compared with the one constructed with their respective CDS to obtain the K score. For each column, the box represents the second (Q2) and third (Q3) quartiles of the data distribution, the line within the box marks the median, whiskers denote the most extreme data. ANOVA one-way analysis of variance and the *p*-value of Tukey’s test were used to verify the existence of statistical differences between the K-tree scores of the different regions evaluated. * *p*-values < 0.05, ** *p*-values between 0.01 and 0.001, *** *p*-values < 0.001.

**Figure 4 viruses-16-00926-f004:**
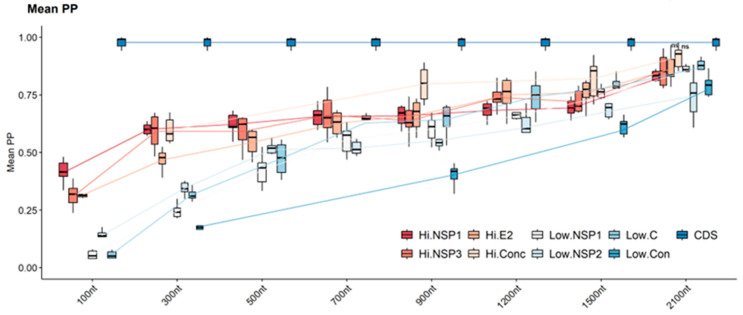
Comparison of tree confidence between CDS tree and trees constructed with high- and low-variation sequences. The trees constructed with the different regions (Hi- and Low-variability) and different lengths (100, 300, 500, 700, 900, 1200, 1500, and 2100 nt) were compared with those constructed with their respective CDS to obtain the mean PP. For each column, the box represents the second (Q2) and third (Q3) quartiles of the data distribution, the line within the box marks the median, whiskers denote the most extreme data. ANOVA one-way analysis of variance and the *p*-value of Tukey’s test were used to identify specific regions with non-significant differences compared to the CDS phylogeny. Regions with *p*-values greater than 0.05 are indicated by “ns”.

**Figure 5 viruses-16-00926-f005:**
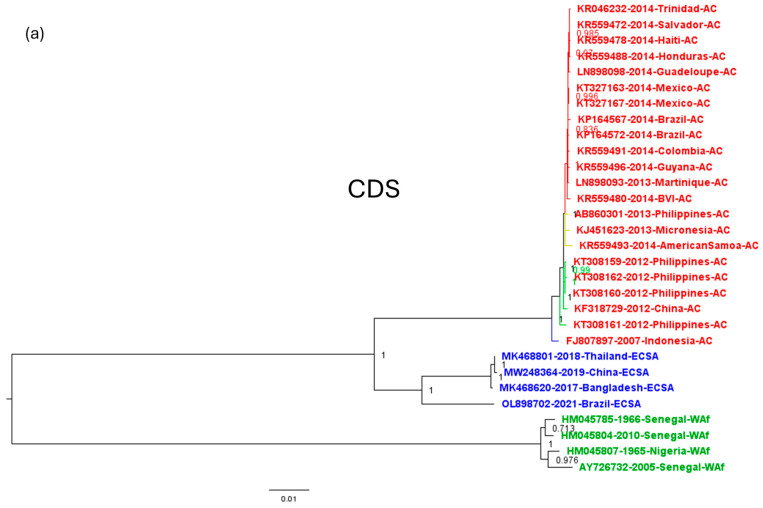
Comparison of the accuracy of CDS trees and those constructed with the Hi-NSP1 region and the concatenated highly variable regions. The trees constructed with the different regions (Hi- and Low-variables) of 2100 nt length were compared with those constructed with their respective CDS trees to assess the congruence between the evolutionary relationships inferred from these distinct genotypes and geographic regions. Only the (**b**) Hi-NSP1 and (**c**) the three highly variable regions were able to adequately infer the evolutionary history of the American outbreak sequences obtained with the (**a**) CDS. The labels are color-coded to represent the three CHIKV genotypes: West Africa (Blue), ECSA (Green), and Asian and Caribbean (Red). Tree branches are colored to indicate lineages within the Asian and Caribbean genotype: Asian lineage from 2007 (Blue), Asian lineage prior to the 2012 outbreak (Green), Asian lineage with the adaptive mutations for dispersal in *Ae. Aegypti* (Yellow), and Caribbean lineage (Red).

## Data Availability

The data presented in this study are openly available in the Bacterial and Viral Bioinformatics Resource Center (BV-BRC) of the National Institute of Allergy and Infectious Diseases (NIH/DHHS) available at https://www.bv-brc.org/ (accessed on 5 February 2024).

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
