# Peer review of "Accurate Recapitulation of Chikungunya Virus Complete Coding Sequence Phylogeny Using Variable Genome Regions for Genomic Surveillance"

_viruses, 2024, doi:10.3390/v16060926_

Round 1

Reviewer 1 Report

Comments and Suggestions for Authors

The analysis by Rodríguez-Aguilar et al. is highly suggestive, as it proposes a novel analytical method for whole genome analysis, which necessitates a significant financial investment. Moreover, they propose a method that can be utilized for flaviviruses in general, which represents a highly valuable contribution.

However, it is necessary to differentiate the proposed method from existing methods in the following points.

In flaviviruses, including chikungunya viruses, the most commonly reported genetic information is the envelope gene sequence. In general, envelope genes often yield results that approximate those obtained using full-length CDS. While a higher resolution analysis can be performed using full-length CDS, in many cases, envelope genes are sufficient for a primary analysis. The authors should demonstrate the advantages of utilizing variable regions for this particular aspect of the analysis, accompanied by the underlying data.

Conversely, if a comparison is required for a very short region, the authors should demonstrate the distinction between the analysis conducted using the 3' untranslated region, which is highly diverse and susceptible to alterations over a brief period of time. However, the 3' untranslated region is associated with several drawbacks and is not typically employed for this analysis. Further discussion of the relative merits and drawbacks of the authors' proposed method in comparison to established techniques for analyzing very short-term changes is encouraged.

In order for the authors' method for phylogenetic analysis using a region containing a highly variable region to be widely used, it is necessary to propose primers that efficiently amplify the proposed region. In this study, Authors focus on the theoretical aspects of the method. However, for the authors' method to be widely used, I believe that information on how to obtain the regions used in the analysis is also necessary. Since it is assumed that conserved regions are also detected in this analysis, I propose that they be presented in the paper.

Comments on the Quality of English Language

Several careless errors are apparent. I think that they should be checked by native speakers.

Reviewer 2 Report

Comments and Suggestions for Authors

Rodriguez-Aguilar et al. studied the rate of variability of the Chikugunya
genome using the sliding window method.
They chose 3 areas with high mutation rates, 3 areas with low mutation rates and
they studied the ability of these areas to model the evolution of strains by
comparing the results to similar analyzes carried out on the entire genome.
The manuscript is clear, the methodology correctly described and the results
almost correctly presented.

My comments :

line 31 : You write: "The genome includes...". I would prefer that you talk
about mRNA because, it seems to me, the encapsidated genome contains neither cap
nor polyA tail.

Line 89 : The bibliographic reference [31] is far too old to reflect the price
of sequencing (2010!)

Figure 1 : In my PDF version I don't have a red line for the y-left axis, I have
the blue bold (y-right) line and a black fine line probably for this axis.
You name this axis “Syn and Non-syn mutation rate” but you give percentages? For
me a rate is not a percentage, it is expressed around the value 1.
Why not just name your axis "% of non-syn mutations".

Figure 5 (line 361) : Figure 5 is named Figure 4 in my PDF.

Line 379 : You use the word prohibitive to talk about sequencing cost. Even if
the discussion is already very long (too long?) I would have liked to have a
quantitative comparison taking into account the latest prices of flongle
nanopores (among other things).
